# Physicochemical and Mechanical Performance of Freestanding Boron-Doped Diamond Nanosheets Coated with C:H:N:O Plasma Polymer

**DOI:** 10.3390/ma13081861

**Published:** 2020-04-15

**Authors:** Michał Rycewicz, Łukasz Macewicz, Jiri Kratochvil, Alicja Stanisławska, Mateusz Ficek, Mirosław Sawczak, Vitezslav Stranak, Marek Szkodo, Robert Bogdanowicz

**Affiliations:** 1Department of Metrology and Optoelectronics, Faculty of Electronics, Telecommunication and Informatics, Gdańsk University of Technology, Gabriela Narutowicza 11/12, 80-233 Gdańsk, Poland; micrycew@student.pg.edu.pl (M.R.); lmacewicz@gmail.com (Ł.M.); matficek@pg.edu.pl (M.F.); 2Institute of Physics, Faculty of Science, University of South Bohemia, Branisovska 1760, 37005 Ceske Budejovice, Czech Republic; jkratochvil@prf.jcu.cz (J.K.); stranak@prf.jcu.cz (V.S.); 3Faculty of Mechanical Engineering, Gdańsk University of Technology, Gabriela Narutowicza 11/12, 80-233 Gdańsk, Poland; alicja.stanislawska@pg.edu.pl (A.S.); marek.szkodo@pg.edu.pl (M.S.); 4Polish Academy of Sciences, The Szewalski Institute of Fluid-Flow Machinery, Fiszera 14, 80-231 Gdańsk, Poland; miroslaw.sawczak@imp.gda.pl

**Keywords:** diamond composite, nylon 6.6, magnetron sputtering, nanoindentation, optical constants

## Abstract

The physicochemical and mechanical properties of thin and freestanding heavy boron-doped diamond (BDD) nanosheets coated with a thin C:H:N:O plasma polymer were studied. First, diamond nanosheets were grown and doped with boron on a Ta substrate using the microwave plasma-enhanced chemical vapor deposition technique (MPECVD). Next, the BDD/Ta samples were covered with nylon 6.6 to improve their stability in harsh environments and flexibility during elastic deformations. Plasma polymer films with a thickness of the 500–1000 nm were obtained by magnetron sputtering of a bulk target of nylon 6.6. Hydrophilic nitrogen-rich C:H:N:O was prepared by the sputtering of nylon 6.6. C:H:N:O as a film with high surface energy improves adhesion in ambient conditions. The nylon–diamond interface was perfectly formed, and hence, the adhesion behavior could be attributed to the dissipation of viscoelastic energy originating from irreversible energy loss in soft polymer structure. Diamond surface heterogeneities have been shown to pin the contact edge, indicating that the retraction process causes instantaneous fluctuations on the surface in specified microscale regions. The observed Raman bands at 390, 275, and 220 cm^−1^ were weak; therefore, the obtained films exhibited a low level of nylon 6 polymerization and short-distance arrangement, indicating crystal symmetry and interchain interactions. The mechanical properties of the nylon-on-diamond were determined by a nanoindentation test in multiload mode. Increasing the maximum load during the nanoindentation test resulted in a decreased hardness of the fabricated structure. The integration of freestanding diamond nanosheets will make it possible to design flexible chemical multielectrode sensors.

## 1. Introduction

Composites of semiconductors with polymers have drawn a lot of attention due to their improvemed mechanical [1], optical [2], and chemical [3] properties, including new properties arising from the mixing of phases at the nanometric scale, distinguishing them from their 3D counterparts [4,5]. Since the first exfoliated graphene flake, the field of thin-layered materials and their possible applications have increased significantly, resulting in numerous papers on semiconducting nanomaterials including phosphorene [6], germanene [7], silicene [8], and carbon materials such as graphene [9], nanowall [10], and nanodiamond [11]. Nevertheless, the incorporation of nanoscale materials into various, well-defined architectures is still challenging. The majority of nanomaterials are fabricated with the requirement of a supporting substrate [12], which creates many obstacles in their integration into nanomaterial-based devices [13]. The design of these nanoscale devices demands the development of flexible freestanding nanostructures, providing additional accessibility to multidimensional interactions.

Diamond, among other carbon materials, is recognized for its excellent mechanical [14] and optical [15] properties, and outstanding biocompatibility [16]. Even though pristine diamond is electrically insulating, it can become a semiconductor by incorporating dopants into its structure [17]. One of the most commonly used dopants is boron, changing diamond into a p-type semiconductor, and allowing diamond-based structures to be an important component in electronic circuits [18]. In addition to their extraordinary chemical stability, diamond films, such as nanocrystalline thin films, can be obtained from a number of bottom-up techniques [19,20,21] such as chemical vapor deposition [22], enabling their growth over large areas (e.g., Si wafers of 100 mm in diameter [23]) of nondiamond substrates, and substantially reducing production costs.

Nylons, like many polymers, are characterized by good flexibility [23] and thermal stability [24]. The greatest restriction to their possible application is their moisture sensitivity [25], which limits their physical and transport properties [26]. Therefore, the mechanical behavior of polymers is a crucial property that should be analyzed prior to usage [27]. Essentially, the yield stress decreases with increasing moisture content [28]. Above 50 °C, the C:H:N:O film changes its thickness and properties. Even after 10 min of heating at 200 °C, the thickness is reduced by 1/3, so there can still be a stabilization effect. This problem can be overcome by reinforcing nylon with diamonds by electrospinning [29] or surface functionalization [30], which may offer substantial improvements in the mechanical properties. There have been reports of two-fold increases in hardness and four-fold increases in Young’s Modulus for diamond-polyamide composites [1], which opens the way for new applications such as functionally-engineered thermoplastics [31].

Plasma polymers are highly cross-linked flexible structures which are usually fabricated by PECVD methods [32]. An alternative method is polymeric target magnetron sputtering [33]. Polymers from PTFE [34], polyethylene [35], and nylon [36] targets have already been prepared by this method in previous studies. Plasma polymers usually copy the topography of the underlying surface; therefore, the material is ideal for nanostructure stabilization, as was shown with nanoparticles [37]. Plasma polymers can be further tailored by changing the deposition conditions with regards to the water stability, the presence of functional groups, and surface energy. Sputtered nylon 6,6 was chosen because it has good stability in water if sputtered in argon, and relatively high surface energy, i.e., about 46 mJ/m^2^ [38], which, together with its flexibility, lead to good adhesion on in general, and specifically, on nanomaterials.

In this study, we present a fabrication method of C:N:H:O (nylon-like) coated, thin, freestanding, boron-doped diamond nanosheets and discuss their physicochemical and mechanical performance. The diamond nanosheets were produced with a microwave plasma-assisted chemical vapor deposition (MWPACVD) technique, and nylon-coated with a sputtering system. The optical properties of the nylon were measured using ellipsometry. The morphology and chemical structure were characterized by scanning electron microscopy and Raman spectroscopy, respectively. The mechanical properties of the nanosheets were also studied in view of possible applications in flexible biosensing optical and electrochemical devices.

## 2. Materials and Methods

### 2.1. Diamond Growth

Boron-doped diamonds were deposited on polished tantalum foil (Sigma-Aldrich, Germany, 1 cm × 1 cm × 0.025 mm, 99.9+% metal basis) in an MW PACVD system (SEKI Technotron AX5400S, Japan) with a frequency of 2.45 GHz. Prior to growth, the substrates were seeded with a colloid of nanoscale diamond particles in water for 30 min. The substrate temperature was kept at 500 °C during the deposition process. The plasma microwave power, optimized for diamond growth, was kept at 1100 W. All samples were doped using diborane (B_2_H_6_) as the dopant precursor; the [B]/[C] ratio was 10,000 ppm. Boron-doped diamond growth was performed with a methane concentration below 2% and 300 sccm total flow rate. The pressure inside the chamber, adjusted by a gas controller, was kept at 50 Torr. The growth time was 720 min, which resulted in polycrystalline foil with a thickness of approximately 4.2 µm. The charge carrier density of the fabricated boron-doped diamond nanosheet was 6.2 × 10^19^ cm^−3^, while the Hall mobility was 9 cm^2^ V^−1^ s^−1^. The Hall effect and resistivity measurements were carried out by the Van der Pauw method at room temperature with the 0.55 T Hall effect setup (HMS-3000, Ecopia, Gyeongsu-daero, Korea).

### 2.2. Deposition of Polymeric C:N:H:O (Nylon-like) Films

Plasma polymeric C:N:H:O (nylon-like) films were deposited using a 3-inch balanced magnetron equipped with a nylon 6.6 target (3 mm thickness, Goodfellow, Germany); see Figure 1. First, the chamber was pumped by a diffusion pump (Balzers PDI160-w, Germany) to an pressure lower than 5 × 10^−4^ Pa, which was measured by a pressure gauge (Pfeifer PKR160-w, Poland). Then, 20 SCCM of argon was introduced into the chamber through a flow controller (MKS 1179A, USA). A pressure of 3 Pa was then set by the positioning of the desk valve which is in-between the chamber and pumping system. After pressure stabilization, the discharge was ignited by delivering 50 W of power to the dielectric target by a radio-frequency generator (Comet Cito 1360-ACNA-P37A-FF, USA) working at a frequency of 13.56 MHz. After 15 min of discharge stabilization, the samples were inserted by the load-lock system into the chamber at a distance 6 cm from the magnetron, which was given by chamber geometry, so that the temperature of the substrate did not exceed 50 °C. The deposition was stopped by taking the samples out and placing them back in the load-lock chamber, where they were left for 15 min under a vacuum in order to let the plasma polymer relax, which improves the film stability. These deposition conditions were selected based on our previous studies [34] and [36]. The deposition speed was estimated using spectroscopic ellipsometry as (8.4 ± 0.8) nm·min^−1^.

To measure the mechanical properties of the nylon-coated, boron-doped, diamond nanosheet, we transferred it to a p-type silicon substrate. The procedure is shown in Figure 2. The fabricated nanosheets showed low adhesion to the tantalum substrate. As a result, they could be mechanically removed (via tweezers) from the substrate and transferred to silver paste (EPO-TEK H20E, Epoxy Technology, USA), which had been previously deposited on a silicon substrate. Next, the sample was cured for 3 h in a vacuum oven (DZ-2BC II, Chemland, Poland) at 80 °C. For our studies, we decided to use a homogenous diamond nanosheet of approximately 1.5 mm × 1 mm in size.

### 2.3. Surface Morphology

In order to define the film morphology, a scanning electron microscope (S-3400 N, HITACHI, Japan) with a tungsten source and variable chamber pressure (VP-SEM) was utilized. Photographs were taken with a (C-5060, Olympus, Japan) camera and a (SZ-630T, Delta Optical, Poland) microscope for three samples.

### 2.4. Raman Spectroscopy

The composition of the deposited films was analyzed by Raman spectroscopy using a micro Raman spectrometer (InVia, Renishaw, UK). Spectra were recorded in the range of 140–3200 cm^−1^ with an integration time of 5 s (10 averages) using a semiconductor laser (785 nm) as an excitation source. Samples were measured in three places (on each of two samples), and the most representative results are presented.

### 2.5. Spectroscopic Ellipsometry

Spectroscopic ellipsometry was measured (two samples) in variable angle mode (55°, 65°, 75°) at wavelengths in the range 210–1690 nm (Woollam M-2000 XI-210, USA). The spectra were modelled by two layers, the plasma polymer film bulk was modelled using one Tauc-Lorentz oscillator, and the topmost film was modelled by the same oscillator, but in the mix with 50% void using Bruggeman effective medium approximation (EMA) to get the surface roughness.

### 2.6. Nanoindentation Test

Nanoindentation tests were performed with a NanoTest™ Vantage (Micro Materials) using a Berkovich three-sided pyramidal diamond. Measurements were made on two samples on which 500- and 2000-nm-thick nylon coatings were deposited. A diamond reference film without nylon coating was also used as the reference sample. On each sample, ten measurements with a multiple load cycle were performed. The indentation with increasing indenter displacement contained 5–10 cycles. For the 500-nm-thick nylon coating, the indenter displacement depth was set in the range from 650 nm to 1450 nm to determine the hardness in the transition zone on the border of the nylon coating and the diamond substrate. Additionally, in order to determine the mechanical properties of the nylon, 25 hardness measurements with a single load cycle were performed on a 500-nm-thick nylon coating with an indenter displacement of 280 nm. For the 2000-nm-thick nylon coating, the indenter displacement depth was set in the range from 550 nm to 2300 nm to determine the homogeneity of the nylon coating. The loading and unloading rate was 0.5 mN/s, and 5 s dwell at maximum load was applied. Based on the load-displacement curves and the Oliver and Pharr method, the surface hardness (H), and reduced Young’s modulus (E) were calculated using the integrated software.

## 3. Results

### 3.1. Topography of Nylon and Stability in Water

The roughness of the 489 ± 11-nm-thick C:H:N:O film deposited separately on a Si wafer was estimated to be 6.75 nm by spectroscopic ellipsometry. Such a low value of surface roughness highlights the very smooth characteristic of the C:H:N:O film, and since the sputtering process was performed under a relatively high pressure of 3 Pa and low power, it was assumed that such a film would probably copy the surface topography of the underlying diamond foil and not destroy it. The adhesion of the carbon-containing C:H:N:O film to the diamond as a carbon structure can be expected to be high, because of both (i) the possible activation of the nanodiamond surface by the plasma [37], which may lead to the creation of covalent bonds between the nanodiamond and the C:H:N:O film, and (ii) the relatively high surface energy of the C:H:N:O material, i.e., 46 mJ/m^2^, composed of a polar part of 24 mJ/m^2^ and a dispersive part of 21 mJ/m^2^, as measured in our previous study [36], which can lead to electrostatic and dispersive interaction between both materials. Another important parameter is the stability of C:H:N:O in water-based environments. As reported in [36], we found that C:H:N:O films swell by about 17% after 2 h of water immersion, but after full drying of the film, the thickness returns to its original value, which points to the good stability of such a polymer network in water. In other words, C:H:N:O films do not dissolve in water, which is important for applications in water-based environments.

### 3.2. Optical Properties of Nylon Films

The optical properties were measured by spectroscopic ellipsometry. Figure 3A shows the ellipsometer output, with quite a good match of the model, indicating the vertical homogeneity of the 500-nm-thick film and the relevance of the resulting extinction coefficient and refractive index, as shown in Figure 3B. The extinction coefficient was almost zero in the range of 500 nm to 1690 nm, so the sample was transparent almost in the whole visible and near-infrared region. The refractive index, i.e., at 589 nm of film, n_D_ = 1.62, was higher than the value of the sputtering target, i.e., 1.52 (bought from Goodfellow), which indicates the higher crosslinking of the plasma polymer making a denser structure than the sputtering target [39].

### 3.3. Morphology and Raman Spectroscopy of Nylon-coated, Boron-doped Diamond Nanosheet

The surface morphology of bare and nylon-coated diamond nanosheets was investigated using the SEM technique. Camera pictures of both bare and nylon-coated samples are presented in Figure 4A1,B1, respectively. In these photographs, the beginnings of the delamination process can be observed, which facilitates the fabrication of freestanding nanosheets. Figure 4A2,A3 shows the surface of boron-doped diamond nanosheets with diamond crystallites covering the whole captured area. It has been previously reported that the crystallite dimensions vary with different [B]/[C] ratios [40]; in this case, they were up to 500 nm. SEM images of the nylon-coated diamond nanosheets (Figure 4B2,B3) confirmed that the coating process enabled full surface encapsulation with visible changes in the surface morphology. The rough nanodiamond surface (Figure 4A3) caused the uneven deposition of the nylon. The obstructed grains were not fully coated with the C:H:N:O plasma polymer. As a result, the isotropic cracks on the nylon-coated boron doped diamond occurred during the curing process (Figure 4B3).

The Raman spectra recorded for the bare and nylon-coated, boron-doped diamond foils are presented in Figure 5. In the spectrum of the uncoated diamond film, a number of characteristic wide bands centered at 488, 1215, 1334, and 1525 cm^−1^ can be distinguished.

The wide asymmetric bands with maxima near 488 cm^−1^, along with the band located near 1215 cm^−1^, originate from the boron defects; their appearance can be observed at a boron doping level ≥10 k ppm. The band located at 1215 cm^−1^ overlapped with the diamond band centered in the range of wavenumbers from 1290 to 1327 cm^−1^, depending of the boron doping level. The bands located near 1334 cm^−1^ were assigned to sp^2^ amorphous carbon (D band) and 1525 cm^−1^ (G band) respectively.

The Raman spectra recorded for the nylon-coated samples differed significantly from the Raman spectra of the bulk nylon, which indicates that the deposited films were not chemically identical to the nylon target. However, weak Raman bands could be recognized in the spectra recorded for polymer-coated samples, and confirmed the presence of characteristic chemical bonds in the nylon structure, as revealed in detail in the Discussion section. The first was a band centered at 953 cm^−1^ assigned to C–CO stretching mode, making the nylon distinguishable from nylon 6.12 and nylon 6. The second was a peak at 1130 cm^−1^ associated with C–C skeletal backbone stretches, and finally, the bands at 880 cm^−1^, 1300 cm^−1^, and 1440 cm^−1^ were due to CH_2_ rocking, twisting, and banding respectively [41]. The low intensity of the Raman signal was due to the strong fluorescence of the deposited film under laser excitation, but also confirmed the random structure of the plasma-deposited polymer material, as reported by other authors [42].

### 3.4. Mechanical Properties of Nylon-on-Diamond Stack

Figure 6 shows examples of the curves obtained during the nanoindentation test in multiload mode. The hysteresis loop for each sample was recorded, indicating their ability to dissipate energy elastically [43]. Based on the obtained load–unload curves, it was possible to determine the hardness distribution profiles and reduced Young’s modulus for the diamond nanosheet and for the nylon coating of varying thickness deposited on the diamond substrate (see Figure 7, Figure 8 and Figure 9). The hardness distribution profiles and reduced Young’s modulus were obtained on the basis of average values obtained over ten measurements. The error bars are marked in the figures as the standard deviation from the average value. As seen in Figure 7, Figure 8 and Figure 9, a greater dispersion of the results occurred in the case of measurements with smaller indenter displacements. As shown in Figure 9, there was a transition zone between the nylon coating and the diamond substrate. The hardness and reduced Young’s modulus of this transition zone were much higher than for a nylon coating, and much lower than for a diamond substrate. Figure 10 shows the microstructure of a cross-section of the tested samples obtained using SEM. As shown in Figure 10, the thickness of the transition zone was about 1000 nm.

## 4. Discussion

In general, the integration of freestanding diamond nanosheets makes it possible to design flexible chemical multielectrode sensors, which could be stable in the aqueous conditions with wide potential windows and low background currents from the diamond, along with mechanical flexibility thanks to the polymer. The transfer of boron-doped diamond (BDD) onto a thin Parylene-C substrate was reported by Fan et al. [44], demonstrating uniformity, significant adhesion, and high yield. The use of polynorbornene-based polymer as a flexible diamond base enables spin castability, photodefinability, and flexibility, and improved adhesion [45] and moisture resistance. Such a flexible structure could be utilized as perfectly conformed interfaces to the soft biomatter, reducing the forces of intracortical probe electrodes in comparison to the stiff silicon-based approach [46].

The studied nylon coating approach enabled us to achieve comparable flexibility and, with optical indices, a low rate of reflection and interference contrast to function as optically transparent conductive probes. It could serve as a flexible base [38], but, due to its low UV absorption edge and zero extinction coefficient in the VIS and NIR range, it will be applied as a spectroelectrochemical sensing surface with large anisotropy, providing not only intensity-resolved interaction, but also phase-sensing polarization data.

Even though nylon covered most of the captured area, many submicron cracks could be seen. These cracks were induced by the growth mechanism on the nanodiamond during deposition because such topography did not match the ellipsometry data for the flat film deposited on the Si wafer, where the roughness of the nylon film was estimated to be 6.75 nm, which points to a dense film with a small portion of void present in only the topmost layer. These cracks were influenced by the nanodiamond substrate beneath the film. The cracks were isotropic, which is typical for films thicker than the size of the grains on the substrate. An analogous effect was reported by Tsubone et al. [3], where analogous fracture behavior was achieved for polymer–DLC composites. It was found here, and previously, that the Young’s modulus plays a critical role in both the cracks and adhesion.

The studied diamond nanosheet demonstrated a polycrystalline nature dominated by (111) facet configuration accompanied with minor contributions of 220 and 311 [47]. DFT-based, ab initio studies of the surface energies of the (100), (111), and (110) diamond faces revealed that diamond exhibited large variations in the range of 4.8–12 J/m^2^, depending on the facet type and surface reconstruction [48].

The (111) face with one dangling bond per surface atoms underwent a strong 2 × 1 reconstruction, called Pandey-chain reconstruction, where conformation allowed a delocalized π bond to form along the surface atoms, thereby stabilizing the surface. The arrangement of the lower lying atoms was a consequence of the constraints imposed by the ‘zig-zag’ chains. Another DFT calculation report revealed that terminating the diamond surface with hydrogen produces lower surface energies than termination by oxygen or hydroxyl groups [49]. The adhesion of the polymer on the nanosheet surface can be reduced by causing the surfaces to be terminated with –H, as opposed to terminating with –OH or =O.

The experimental surface energy of polyamide Nylon-6,6 reached 46 mJ/m^2^. Our previous studies of oxygen-terminated, boron-doped diamond resulted in an average surface energy 53.4 mJ/m^2^ through contact angle experiments of 40 deg [50]. The hydrogenated surface exhibited reduced surface energy at contact angles of 80°, becoming more hydrophobic [51]. The strength of the adhesive, contact, nylon-diamond nanosheet was determined by measuring the relative surface energies of the two surfaces [52]. We applied an oxygen-terminated diamond surface to reduce the relative surface energy and enhance the nylon adhesion.

Nevertheless, the deposited polycrystalline diamond nanosheet was neither homogeneous nor entirely oxygen terminated. Most of the (111) faces were oxidized, thereby achieving nylon adhesion, while other facets and intragrain regions were still predominantly hydrogen terminated and rich in CH defects. Diamond surface heterogeneities have been shown to pin the contact edge, indicating that the retraction process deepens the surface in instantaneous fluctuations over specified microscale regions [53]. Such a region results in larger surface energies related to weak interface interactions manifesting as cracks and valleys in the SEM images. The interface nylon formed complete contact; hence, its adhesion behavior was attributed to the viscoelastic energy dissipation originating from an irreversible energy loss in the soft polymer structure. Its roughness-driven behavior was dependent on the surface area and Griffith-like separation of the interface, and cannot be quantitatively linked to surface topography [53].

Despite many submicron cracks, the C:H:N:O-coated, boron-doped diamond nanosheet did not conduct an electric current in the sandwich configuration. Before deposition of nylon, freestanding, boron-doped, diamond nanosheets showed a resistivity of 0.11 mΩ·cm and had a hole concentration of 6.2 × 10^19^ cm^−3^.

The recorded Raman bands of the nylon-coated, boron-doped diamond foils were very broad and weak, which made their identification challenging. The broadening of the Raman bands indicated a decrease of the material crystallinity, and pointed to a more random structure. Such behavior is common for plasma-deposited polymer films, and reported by other authors [42,54].

The Raman data also revealed weak bands at 275, and 220 cm^−1^, which were attributed to the calculated values at 273, and ∼210 cm^−1^ reported by Yamamoto et al. [55]. The ~220 cm^−1^ band was attributed to the torsional vibration of methylene groups and the out-of-plane motion of the NH group. This is characteristic of the α form of nylon 6. The distortion of C−CH_2_−CH_2_ and the bending motion of the carbonyl groups in the amide plane were connected to the 275 cm^−1^ band. Since the bands were weak and degenerated, the obtained films exhibited a low level of nylon 6 polymerization and short-distance arrangement, manifesting crystal symmetry and interchain interactions. These findings indicate the mechanical properties, water resistance, and low optical absorption coefficient.

A previous study of nylon-carbon nanotube interactions showed direct covalent bonds between the nanotubes and nylon chains via amide linkages or via alkyl segments, depending on the nanotube orientation [56]. This phenomena could also take place in multifaceted diamond nanosheets. It was reported that amide binding reduces nylon flexibility, while providing improved chain flexibility [57]. These mutlibinding mechanism may also be responsible for film stresses, shifted Raman bands, and morphologic defects.

The measured hardness of the diamond surface layer differed depending on the load applied to the Berkovich indenter. As shown in Figure 7, the indentation size effect (ISE) can be observed in the indentation through testing at different indenter loads. Increasing the maximum load during the nanoindentation test resulted in a decrease in hardness. According to Nix and Gao [58], the ISE is directly related to geometrically necessary dislocations (GNDs) whose density is proportional to the inverse of the indentation depth. In the initial stage of the nanoindentation test, GNDs were generated in the tested material to conform to the shape of the indenter and enable the crystal rotation. These dislocations were stored in the plasticized material and they interact with the dislocations generated during the diamond depositing process (i.e., statistically-stored dislocations, SSDs). Prior molecular dynamics simulations of polycrystalline diamond revealed that two plastic deformation modes induced during deposition might be attributed to the dislocation propagation mode and the atomic disordering mode [59]. Furthermore, Huang et al. [60] reported that the strain rate exhibits a minor effect on the Young’s modulus of the diamond surfaces. This could alter the inelastic and fracture effects, but its general influence is reduced and compensated for by the redistribution and relaxation of local residual stresses between the crystalline columns. For a cone-shaped indenter, the density of the GNDs is obtained as [61]:(1)$$\rho_{GND}=\frac{3}{2}\cdot \frac{1}{f^{3}}\cdot \frac{tan^{2}\theta }{bh}$$
where *θ* is the angle between the surface and the indenter (for a Berkovich indenter, *θ* = 65.35°), *h* is the indentation displacement, *b* is the magnitude of the Burgers vector, and factor *f* = *apz*/*ac*, where *apz* is the radius of the plasticized zone during penetration of the nanoindenter, and *ac* is the radius of contact of the indenter with the tested material. Although the GND density analysis described above refers to a cone-shaped indenter, the same relationship between GND density and indenter depth has been obtained for other indenter shapes. As can be seen from Equation (1), GND density, *ρ_GNDs_*, is proportional to the inverse of the indentation depth, and when the indenter displaces for longer distances, the density of the GNDs decreases. Hardness as a function of indentation depth is given by the linear superposition of the GND and SSD density [61]:(2)$$H_{ISE}=MC\alpha Gb\cdot \sqrt{\rho_{SSD}+\rho_{GND}\left(h\right)}$$
where *M* is a Taylor factor, *α* is a factor dependent on the dislocation substructure, *C* is a constraint factor (transferring the complex stress state underneath the indenter in a uniaxial strain field), and *G* is the shear modulus.

The ISE can also be seen for the nylon coating (see Figure 8). For polymers, it is assumed that the molecular interactions related to the Frank energy are important for the deformation process at small length scales because the lack of the nematic order is due to topological defects like cross links and entanglements, rather than to the absence of sufficiently strong molecular interactions. This Frank energy can explain the ISE in polymers as the gradients in the rotations increase with decreasing indentation depths. Based on this assumption, a depth-dependent hardness model *H* = *H*_0_[1 + (*c*/*h*)*^γ^*] was deduced, where *H*_0_ is the macroscopic hardness, *c* is a length scale parameter, *h* is the displacement of the indenter, and *γ* is and fitting parameter to account for the assumption applied in its derivation [62].

Figure 9 shows changes in hardness and reduced Young’s modulus in the transition zone between the nylon coating and the diamond substrate, and Figure 10 shows the appearance of the transition zone. As is apparent from these figures, the mechanical properties in the transition zone had an intermediate value between the properties of the nylon coating and those of the diamond substrate. This was due to the fact that the transition zone was a mixture of nylon and diamond formed during the process of depositing the nylon on the diamond substrate. During this process, the nylon particles initially filled the pores and were deposited on the diamond substrate.

## 5. Conclusions

In summary, the deposition of nylon on a thin, freestanding, boron-doped, diamond sheet results in mechanically-stable stacks. The results of Scanning Electron Microscopy and Raman spectroscopy confirmed that the nylon coating formed a fully encapsulated surface. The extinction coefficient and refractive index gathered with spectroscopic ellipsometry suggested that the fabricated samples were transparent in the majority of the visible and near-infrared region.

Even though nylon covered most of the captured area, many submicron cracks could be seen. Those cracks are probably created by the deposition mechanism at the polycrystalline elastic nanosheet surface. The Raman spectra indicated that the deposited films were not chemically identical to the nylon target. The observed Raman bands were very broad and weak, which indicated a decrease of material crystallinity, and pointed to a more random structure of the polymeric material. Such behavior is common for plasma-deposited polymer films.

The mechanical properties of the nylon-on-diamond stack were obtained during the nanoindentation test in mutliload mode. An increasing maximum load during the nanoindentation test resulted in a decrease in the hardness of the fabricated material. The use of nylon enabled us to achieve a flexible and transparent, freestanding diamond structure, while pristine nanosheets were brittle and fragile. Composite diamond-on-polymer structures could be further developed for flexible and robust electronic sensors or thermal heat spreaders.

## Figures and Tables

**Figure 1 materials-13-01861-f001:**
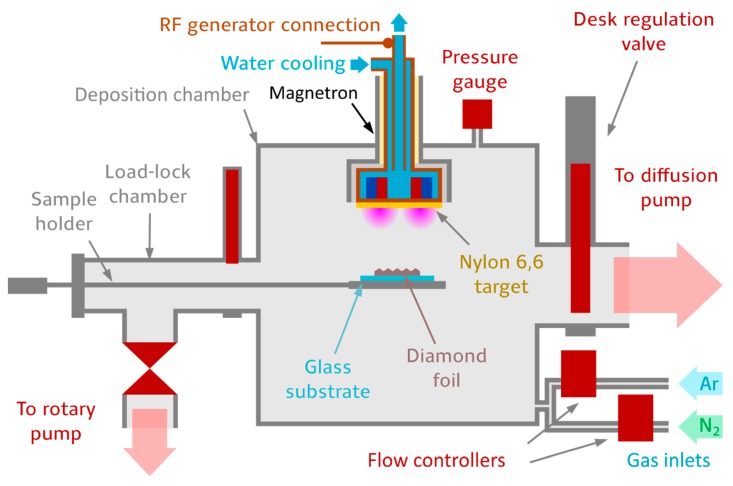
Deposition setup for nanodiamond overcoating with a C:H:N:O film. The load-lock system ensures a high-quality vacuum and good quality thin films because there is no need to vent the chamber before each deposition. The chamber is equipped with a planar 3-inch magnetron, the gas is introduced into the chamber by flow controllers, and the pressure is regulated by a desk regulation valve connecting the chamber and vacuum pump.

**Figure 2 materials-13-01861-f002:**
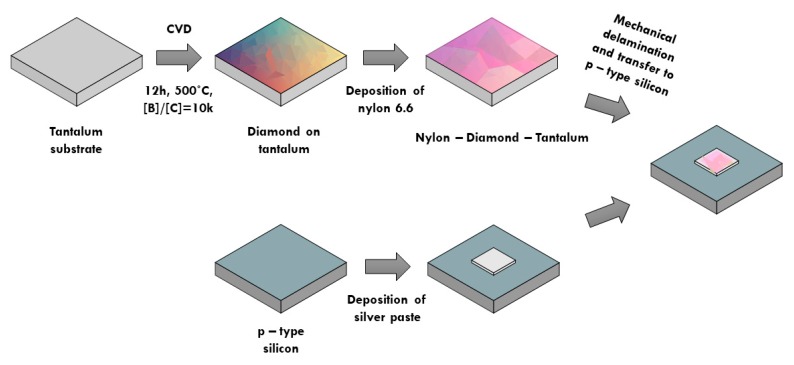
Procedure for the fabrication of nylon-coated, boron-doped, diamond nanosheet samples for nanoindentation test.

**Figure 3 materials-13-01861-f003:**
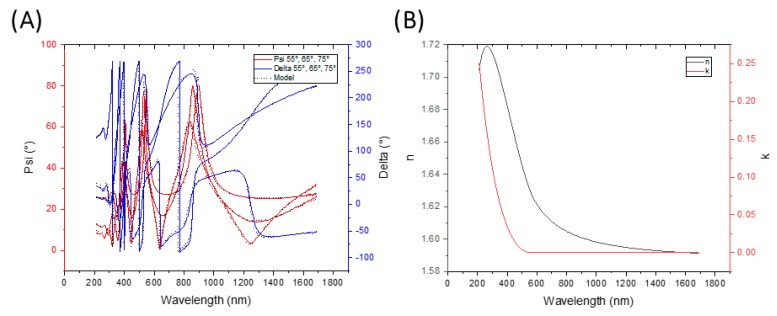
Optical properties of plasma polymerized nylon. (**A**) Modelling of spectroscopic ellipsometry, (**B**) Plots of refractive index and extinction coefficient.

**Figure 4 materials-13-01861-f004:**
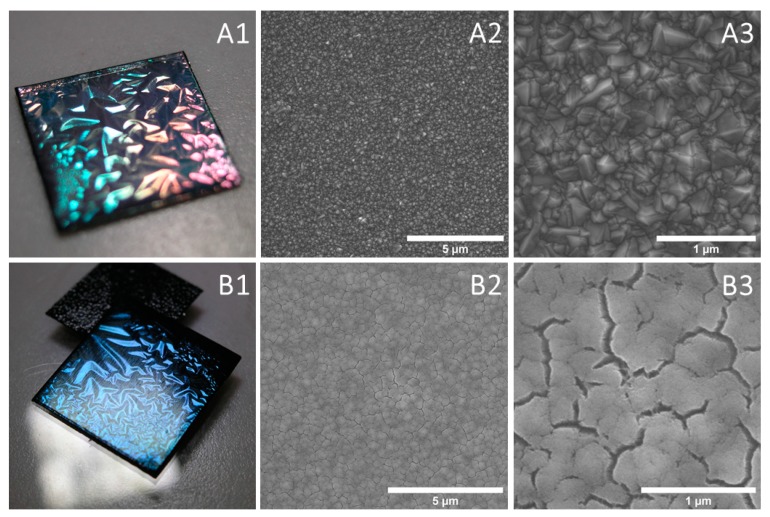
Photographs and SEM images of boron-doped diamond nanosheet on a tantalum substrate (**A1**–**A3**) and coated with nylon (**B1**–**B3**) at various magnifications.

**Figure 5 materials-13-01861-f005:**
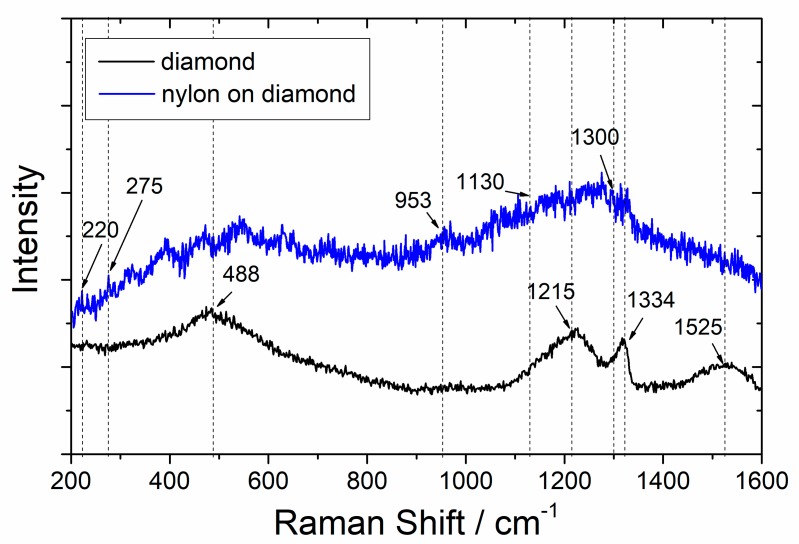
Comparison of Raman spectra for as-grown, boron-doped, diamond and nylon-coated diamond nanosheet.

**Figure 6 materials-13-01861-f006:**
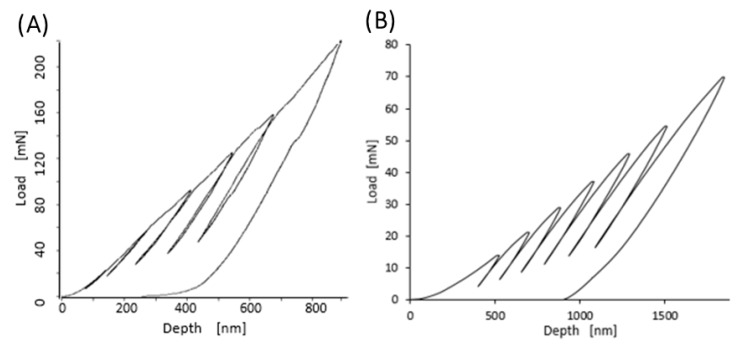
Sample load and unload curves obtained in a multiple-load cycle with an increasing load for diamond nanosheet—(**A**), and for a nylon coating with a thickness of 500 nm—(**B**).

**Figure 7 materials-13-01861-f007:**
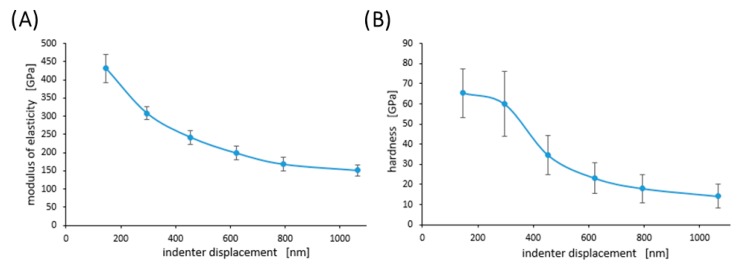
Profiles of hardness (H)—(**A**), and modulus of elasticity (E)—(**B**) for diamond nanosheet. Error bars represent ± standard deviation.

**Figure 8 materials-13-01861-f008:**
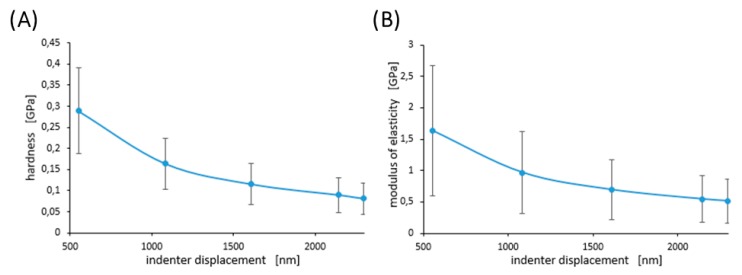
Hardness—(**A**), and modulus of elasticity—(**B**) for a nylon coating with a thickness of 2000 nm. Error bars represent ± standard deviation.

**Figure 9 materials-13-01861-f009:**
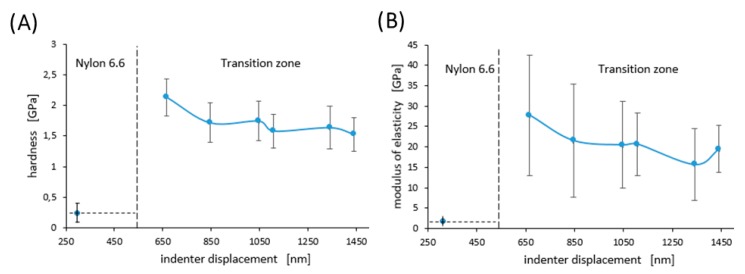
Hardness—(**A**), and modulus of elasticity—(**B**) for a nylon coating with a thickness of 500 nm and for a transition zone. Error bars represent ± standard deviation.

**Figure 10 materials-13-01861-f010:**
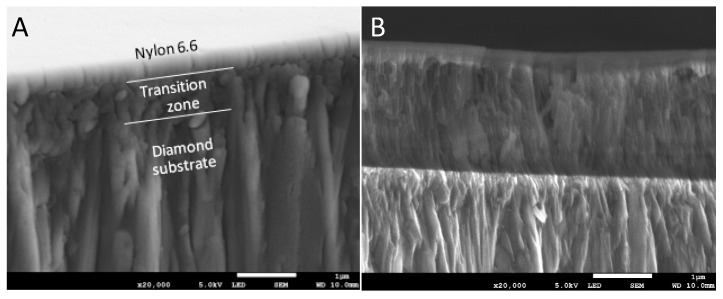
Structure of a cross-section of a 500 nm nylon coating—(**A**), and a 2000 nm nylon coating—(**B**) on a diamond substrate.

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
