# Peer review of "Physicochemical and Mechanical Performance of Freestanding Boron-Doped Diamond Nanosheets Coated with C:H:N:O Plasma Polymer"

_materials, 2020, doi:10.3390/ma13081861_

Round 1

Reviewer 1 Report

The paper deals with the characterization of the free-standing heavy boron-doped diamond (BDD) nanosheets coated by a thin layer of polymer. The topic is related to the strive for advancement in flexible electronic devices which is a novel and interesting area.

In the abstract:
- the goals of the paper should be clearly defined.

- the benefits of the author's approach in the manufacturing of semiconducting nanocomposites should be highlighted as it motivates the relevance of the paper.

In the introduction section:

  • listing of the methods that have been used to carry out your study unnecessary in the introduction section (lines 77-82). It could be replaced by the results that you obtained during your study and how these results fit in the goal you stated.

Besides that, Introduction is sufficient but could be more structured and blocks are more connected to each other in terms of the story to improve readability.

In the materials and methods section:

-what type of magnetron do you use? Is it a balanced or an unbalanced magnetron? What was the substrate temperature during the surface functionalization? Could the cracking of your films occur due to excessive heating? Why exactly this throw distance (6 cm) was chosen? Did you try to deposit your films and extended throw distance? Is it based on previous results? If so, please cite them.

-mechanical delamination and transfer of the film should be further clarified. Was the diamond layer adherent to the tantalum substrate and what do you exactly mean by mechanical delamination.

In the results section:

  • Why roughness of the C:H:N:O film deposited separately on Si was tested for the 489 nm film (could you please use "±" instead of "+/-" )? In the materials and methods section, you stated that the deposition time was 15 mins and deposition speed was estimated to be (8.4 ± 0.8) nm.min-1.
  • Could you please motivate the importance of the fact that "C:H:N:O films do not dissolve in water".
  • What is the reason for color change in boron-doped diamond nanosheet on tantalum substrate visible in the photograph A1, fig. 4? Is it due to the inhomogeneity? How critical it is to your study?
  • Could the cracking appear due to the high surface free energy of the diamond layer? 
  • In Fig. 10 you are showing nylon coating of 500 nm and 2000 nm, however, in the materials and methods section you mention the deposition for only 15 minutes. What is the surface morphology of 2000 nm nylon coating on diamond then? Is it the same as you showed in fig. 4?

In the discussion section:

  • Much attention is given to the hardness measurement. Besides that, discussion seems poor and needs to be improved.

Functional tests of free-standing boron-doped diamond nanosheets in relation to the desired application in the field of electronic devices could be of interest to the reader.

Finally, English must be improved.

Author Response

Please find the response letter in the separated file.

Reviewer 2 Report

In this work, authors reported in depth investigation on free-standing heavy boron-doped diamond nanosheets coated by C:H:N:O plasma polymer. They find out physicochemical and mechanical properties of thin films with reasonable results. Therefore, I recommend its publication, with suggested modifications, as below:

  • What is the p-type concentration of the boron-doped diamond on tantalum with and without nylon layer
  • What is the p-type concentration of the boron-doped diamond before and after transfer on the p-type silicon
  • If possible use the Hall measurement results for p-type concentration studies

Author Response

(The authors gave the same response as above.)

Reviewer 3 Report

General comments

The manuscript concerns the characterization of a C:H:N:O plasma coating. The work is well structured and organized. Unfortunately, you have to improve the quality of the Discussion section that does not appear as an adequate comparison to the literature. Moreover, better highlight the novelty of the materials. Report possible applications in the different industrial fields, reported in in the Introduction. Detailed comments are reported here below.

Detailed comments

- line 57: give a range for “large areas”

- line 59: report a range for flexibility and thermal transition temperatures

- lines 63-65: 1) better describe how nylon can be reinforced with diamond; 2) give a quantitative range of possible improvement in the mechanical properties

- line 66: give a reason for “highly cross-linked”

- line 72: give a range for surface energy

- line 95: it is not clear if you have optimized the process here described or if you get this protocol fro literature

- line 124, line 129, line 134: report the number of replicates used for each characterization method; report the type of samples that you analysed

- line 140: better explain if the measurements were acquired on the same specimen, or if you used different specimens for each type of considered sample

- line 161-164: you have to better analyse this possible explanation; you have to perform adequate chemical analyses to prove your hypothesis

- line 168: you have to report in the Materials and Methods section the protocol used to study the stability in water

- line 171-172: do you have an explanation for this result? you have to add it

- line 194-195: you have to demonstrated by experimental data what you are here describing

- Figurea-B3: it seems that some cracks occurred on the surface. You have to better describe this image

- Figure 5: many peaks (probably due to noisy background) are visible in the Raman spectrum of the nylon-coated sample. You have to better describe the spectrum, and you have to better describe how you could considered the peak here highlighted

- line 217: a deeper comparison between the materials here characterized is suggested

- Discussion: you have to add a deeper comparison with the literature

Author Response

(The authors gave the same response as above.)

Round 2

Reviewer 3 Report

The Authors have taken into consideration the reviewer's comments and the quality of the manuscript has been improved.

No additional modifications are required.